# Ventilator Parameters in the Diagnosis and Prognosis of Acute Respiratory Distress Syndrome in Postoperative Patients: A Preliminary Study

**DOI:** 10.3390/diagnostics11040648

**Published:** 2021-04-03

**Authors:** Chew-Teng Kor, Kai-Huang Lin, Chen-Hsu Wang, Jui-Feng Lin, Cheng-Deng Kuo

**Affiliations:** 1Internal Medicine Research Center, Department of Research, Changhua Christian Hospital, Changhua 500, Taiwan; 179297@cch.org.tw; 2Graduate Institute of Statistics and Information Science, National Changhua University of Education, Changhua 500, Taiwan; 3Division of Critical Care Medicine, Department of Internal Medicine, Changhua Christian Hospital, Changhua 500, Taiwan; 54407@cch.org.tw; 4Medical Intensive Care Unit, Division of Cardiology, Department of Internal Medicine, Cathay General Hospital, Taipei 106, Taiwan; sunnychw@gmail.com; 5Division of Neurosurgery, Department of Surgery, Mackay Memorial Hospital, Taipei 104, Taiwan; ljfgod@yahoo.com.tw; 6Institute of Traditional Medicine, National Yang-Ming Chiao-Tung University School of Medicine, Taipei 112, Taiwan; 7Department of Medical Research, Taipei Veterans General Hospital, Taipei 112, Taiwan; 8Department of Medicine, Taian Hospital, Taipei 104, Taiwan; 9Tanyu Research Laboratory, Taipei 112, Taiwan

**Keywords:** acute respiratory distress syndrome, rapid shallow breath index, work of ventilation, rate pressure product of ventilation, inspiration to expiration time ratio

## Abstract

This study investigated the usefulness of ventilator parameters in the prediction of development and outcome of acute respiratory distress syndrome (ARDS) in postoperative patients with esophageal or lung cancer on admission to the surgical intensive care unit (SICU). A total of 32 post-operative patients with lung or esophageal cancer from SICU in a tertiary medical center were retrospectively analyzed. The study patients were divided into an ARDS group (*n* = 21) and a non-ARDS group (*n* = 11). The ARDS group contained the postoperative patients who developed ARDS after lung or esophageal cancer surgery. The ventilator variables were analyzed in this study. Principal component analysis (PCA) was performed to reduce the correlated ventilator variables to a small set of variables. The top three ventilator variables with large coefficients, as determined by PCA, were considered as sensitive variables and included in the analysis model based on the rule of 10 events per variable. Firth logistic regression with selective stepwise elimination procedure was performed to identify the most important predictors of morbidity and mortality in patients with ARDS. Ventilator parameters, including rapid shallow breath index during mechanical ventilation (RSBIv), rate pressure product of ventilation (RPPv), rate pressure volume index (RPVI), mechanical work (MW), and inspiration to expiration time ratio (IER), were analyzed in this study. It was found that the ARDS patients had significantly greater respiratory rate (RR), airway resistance (Raw), RSBIv, RPPv, RPVI, positive end-expiratory pressure (PEEP), and IER and significantly lower respiratory interval (RI), expiration time (Te), flow rate ($\dot{V}$), tidal volume (V_T_), dynamic compliance (Cdyn), mechanical work of ventilation (MW), and MW/IER ratio than the non-ARDS patients. The non-survivors of ARDS had significantly greater peak inspiratory pressure above PEEP (PIP), RSBIv, RPPv, and RPVI than the survivors of ARDS. By using PCA, the MW/IER was found to be the most important predictor of the development of ARDS, and both RPPv and RPVI were significant predictors of mortality in patients with ARDS. In conclusion, some ventilator parameters, such as RPPv, RPVI, and MW/IER defined in this study, can be derived from ventilator readings and used to predict the development and outcome of ARDS in mechanically ventilated patients on admission to the SICU.

## 1. Introduction

Acute respiratory distress syndrome (ARDS), a life-threatening inflammatory lung disease that affects both medical and surgical patients, was first reported by Ashbaugh and colleagues in 1967 in a case series of 12 ICU patients [1]. The clinical, radiological, biochemical, and pathological features of ARDS were defined in 1994 by the American–European Consensus Conference (AECC) [2]. The current definition of ARDS is the “Berlin Definition” devised by a panel of experts in 2013 based on the timing of clinical insult, radiographic pattern, ratio of partial pressure of arterial oxygen to fraction of inspired oxygen (PaO_2_/FiO_2_), and positive end-expiratory pressure (PEEP) [3]. ARDS is associated with an extremely high mortality rate in patients with lung resection or esophagectomy [4,5,6,7,8].

ARDS is a diffuse, progressive inflammatory lung disease with hypoxemia, reduced lung compliance, and bilateral opacities on chest X-ray image. Although mechanical ventilation provides essential life support, it can also induce and worsen lung injury [9,10]. However, mechanical ventilation strategies remain the mainstay of respiratory support for patients with ARDS. Due to the progress of modern digital technology, advanced new-generation ventilators allow extensive and integrated monitoring of patients’ respiratory mechanics. Data regarding the control variables, phase variables, breath type, respiration rates, tidal volume, and so on can be easily read and constantly recorded from the ventilator.

## 2. Objective

In this study, we investigated the usefulness of ventilator parameters derived from the ventilator readings in the prediction of development and outcome of ARDS in postoperative patients with lung or esophageal cancer on admission to the surgical ICU (SICU). No other clinical parameters, scores, or laboratory data were included in the analysis. We hypothesize that ventilator parameters alone can reflect the disease severity and help diagnose and predict the outcome of the patients with ARDS.

## 3. Material and Methods

### 3.1. Study Design

This was a single-center, case-controlled study with retrospective data analysis. The study protocol was approved by the Institutional Review Boards of National Taiwan University Hospital (NTUH200808065R, approval date: 17 September 2008) and Taipei Veterans General Hospital (VGHIRB97-01-02A, approval date: 21 April 2008). Retrospective analysis of the data to find the key ventilator parameters for the prediction of development and outcome of ARDS in postoperative patients with esophageal or lung cancer was approved by the Institutional Review Board of Changhua Christian Hospital (CCH IRB 190521, approval date: 20 June 2019).

This study was conducted in the SICU of the National Taiwan University Hospital. The patient selection criteria and clinical data have been presented in our previous article [11]. In brief, all patients were over 18 years old and had received thoracic surgery for lung or esophageal cancer. They were transferred to the SICU for postoperative care. Patients without postoperative ARDS were included as the control group, while patients who were complicated with ARDS were included as the ARDS group. The ARDS of the patients was diagnosed according to the Berlin Definition [12]. Three categories of ARDS based on the degree of hypoxemia and four ancillary variables for severe ARDS were adopted. The hypoxemia was classified as mild (200 mmHg < PaO_2_/FIO_2_ ≤ 300 mmHg), moderate (100 mmHg < PaO_2_/FIO_2_ ≤ 200 mmHg), and severe (PaO_2_/FIO_2_ ≤ 100 mmHg). The four ancillary variables for severe ARDS were radiographic severity, respiratory system compliance (≤40 mL/cmH_2_O), positive end-expiratory pressure (≥10 cmH_2_O), and corrected expired volume per minute (≥10 L/min). Patients who had severe coronary artery disease, persistent arrhythmia, cardiac pacing, diabetes mellitus, cerebral vascular accident, or major diseases of kidney or autoimmune system were excluded from the study. Eleven patients in the non-ARDS group and 21 patients in the ARDS group were analyzed in this study.

All patients were intubated and mechanically ventilated using pressure control mode. Fentanyl was given to all patients as an analgesic. The demographic data, vital signs, medications, ventilator readings, and relevant clinical data were recorded within 4 h of admission to the SICU.

### 3.2. Ventilator Parameters

The flow rate ($\dot{V}$) of gas during mechanical ventilation is given by the following equation:(1)$$\dot{V}=\frac{V_{T}}{\mathrm{Ti}},$$
where V_T_ is the tidal volume and Ti is the inspiration time. The pressure control mode of ventilation typically has a decelerating pattern, and V_T_/Ti will provide an average flow rate that depends on both resistance and compliance. The airway resistance (Raw) to airflow during mechanical ventilation is given by the ratio of peak inspiratory pressure above PEEP (PIP) to the flow rate $\dot{V}$:(2)$$\mathrm{Raw}=\frac{\mathrm{PIP}}{\dot{V}}.$$

The dynamic compliance (Cdyn) represents the pulmonary compliance during inspiration and is given by the ratio of tidal volume to the PIP during inspiration:(3)$$\mathrm{Cdyn}=\frac{V_{T}}{\mathrm{PIP}}.$$

The rapid shallow breathing index (RSBI) is thought to be a stress response reflecting the balance between respiratory neuromuscular reserve and respiratory demands, and it has been widely used in daily clinical practice such as in the prediction of successful weaning from ventilator [13,14,15,16]. When the patient is on the ventilator using pressure control or pressure support mode of ventilation, the RSBI during mechanical ventilation (RSBIv) can also be defined as the ratio of respiration rate (RR) to the tidal volume:(4)$$\mathrm{RSBIv}=\frac{\mathrm{RR}}{V_{T}}.$$

The “v” in RSBIv indicates that this RSBIv is measured when the patient is still on the ventilator, which is different from the RSBI measured when the patient is temporarily discontinued from mechanical ventilation.

The product of heart rate (HR) and systolic blood pressure (SBP), or the rate pressure product (RPP), is a very reliable indicator of myocardial oxygen demand and has been widely used clinically, especially in cardiology, anesthesiology, and rehabilitation [17,18,19]. By analogy with the RPP in cardiology, the rate pressure product of ventilation (RPPv) can be defined as the product of RR and PIP in ventilated patients to measure the stress imposed on the respiratory muscle in ventilated patients:(5)RPPv=RR·PIP.

Combining RSBIv and RPPv, we can devise the rate pressure volume index (RPVI) as
(6)$$\mathrm{RPVI}=\mathrm{RSBIv}\cdot \mathrm{PIP}=\frac{\mathrm{RR}\cdot \mathrm{PIP}}{V_{T}}=\frac{\mathrm{RPPv}}{V_{T}}.$$

Though RR appears in both RSBIv and RPPv, it is included in the definition of RPVI only once rather than twice so as not to overemphasize its role in the new index of RPVI.

The respiratory interval (RI) is the averaged time of ventilation that can be obtained from the RR using the following equation:(7)$$\mathrm{RI}=\frac{60\;s}{\mathrm{RR}}.$$

The expiration time (Te) is the difference between RI and Ti:(8)Te=RI−Ti.

The inspiration to expiration time ratio (IER) is then
(9)$$\mathrm{IER}=\frac{\mathrm{Ti}}{\mathrm{Te}}.$$

When the gas flows across a constant cross-section, the mechanical work (MW) done by the ventilator can be defined as
(10)$$\mathrm{MW}=\mathrm{PIP}\cdot V_{T},$$
which is different from the conventional work of breathing work done by the patient when the patient is breathing on his/her own effort. As MW is related to both inspiration and expiration times, a new parameter MW/IER can be constructed to reflect the severity of ARDS:(11)$$\frac{\mathrm{MW}}{\mathrm{IER}}=\left(\mathrm{PIP}\cdot V_{T}\right)\frac{\mathrm{Te}}{\mathrm{Ti}}=\mathrm{PIP}\cdot \dot{V}\cdot \mathrm{Te}.$$

### 3.3. Statistical Analysis

Data are presented as a percentage, median (interquartile range), or mean ± standard deviation. Continuous variables were tested for normal distribution using the Kolmogorov–Smirnov test and were compared between the two groups of patients using Mann–Whitney U test for non-normally distributed data or independent samples *t*-test for normally distributed data. Chi-square test or Fisher’s exact test, when appropriate, was used for the comparisons of categorical data.

Firth logistic regression with selective stepwise elimination procedure was performed to identify the most important predictors of morbidity and mortality of ARDS patients. This method uses the penalized likelihood approach to reduce the parameter estimation bias due to small sample size. However, only three variables can be included in the multivariable analysis based on the rule of 10 events per variable (EPV-10), as in our previous study [11]. The candidate mechanical ventilation measurements included in the multivariable analysis were the significant variables in the univariate analysis of differences between the two groups. Because these mechanical ventilation measurements were highly correlated with one another, principal component analysis (PCA) was performed to reduce the correlated variables to a small set of variables. By using the PCA selection method, variables with large coefficients in each component can be considered as sensitive variables because those variables contain most information from the dataset. In this study, the top three variables were selected as the sensitive variables. Furthermore, receiver operating characteristic (ROC) curves were constructed to assess the predictive performance of important measurements. All statistical analyses were performed using R software (v.i386 3.6.2 has been released on 10 December 2019, https://www.r-project.org/) and the R package logistic for Firth’s penalized likelihood logistic regression. A two-tailed *p* < 0.05 was considered statistically significant.

## 4. Results

Table 1 shows that the ARDS patients had significantly longer SICU stay; significantly greater RR, Raw, RSBIv, RPPv, RPVI, PEEP, and IER; and significantly lower RI, Te, $\dot{V}$, V_T_, Cdyn, MW, and MW/IER than the control patients. Table 2 shows that the non-survivors had significantly greater PIP, RSBIv, RPPv, and RPVI than the survivors of ARDS.

Two components were extracted after the PCA procedure, and the coefficients of variables in each component are summarized in Table 3. Variables with coefficients in bold font in the same column in Table 3 are included in the same component. Using PCA, RSBIv, RPVI, and MW/IER were found to be the sensitive variables in component 1, and $\dot{V}$, MW, and V_T_ were found to be the sensitive variables in component 2. Three models were used to assess the predictors of morbidity of ARDS in multivariable analysis. Model 1a and model 2a were adjusted for three sensitive variables in components 1 and 2, respectively.

After performing a selective stepwise elimination procedure for model 1a and model 2a, both MW/IER (*p* = 0.006) and V_T_ (*p* = 0.033) were shown to be significant predictors of the development of ARDS (Table 4). Model 3a was adjusted for MW/IER and V_T_ and showed that MW/IER was the most important predictor of the development of ARDS (Table 4). As depicted in Figure 1A, the AUCs and 95% confidence intervals (CIs) of PEEP, RSBIv, RPVI, V_T_, and $\dot{V}$ in predicting the morbidity of ARDS were 0.931 (95% CI = 0.85 − 1.00; *p* < 0.001), 0.885 (95% CI = 0.77 − 1.00; *p* < 0.001), 0.857 (95% CI = 0.77 − 1.00; *p* < 0.001), 0.816 (95% CI = 0.67 − 1.00; *p* = 0.004), and 0.801 (95% CI = 0.65 − 1.00; *p* = 0.006), respectively. The AUC of the new parameter MW/IER was found to be 0.944 (95% CI = 0.86 − 1.00; *p* < 0.001).

PIP, RSBIv, RPPv, and RPVI were the significant variables responsible for the differences between the survivors and non-survivors of ARDS (Table 2). Because RSBIv, RPPv, and RPVI are highly correlated with one another, we used three separate models to assess the predictors of mortality of ARDS patients. Model 1b was adjusted for PIP and RSBIv and indicated that none of the parameters in model 1b could significantly affect the mortality of ARDS patients (*p* > 0.10). Model 2b and model 3b were adjusted for RPPv and RPVI, respectively. The results indicated that RPPv and RPVI could marginally significantly affect the mortality of ARDS patients (*p* < 0.10) (Table 4). As depicted in Figure 1B, the AUCs of RPVI, RPPv, RSBI, and PIP in predicting the mortality of ARDS patients were 0.850 (95% CI = 0.68–1.00; *p* = 0.021), 0.838 (95% CI = 0.66–1.00; *p* = 0.026), 0.706 (95% CI = 0.49–1.00; *p* = 0.173), and 0.813 (95% CI = 0.63–1.00; *p* = 0.039), respectively.

## 5. Discussion

Despite significant advances in the understanding and treatment of ARDS in the past 50 years, the mortality rate of ARDS remains high [20]. The risk factors and outcome evaluation of ARDS are still important issues for mortality reduction. Many indices have been developed to predict the risk and outcomes of postoperative ARDS in trauma or surgical patients [6,8,21,22,23,24]. However, these indices or scoring systems require the inclusion of a broad range of risk factors and risk modifiers that are difficult to obtain in clinical practice. It is necessary to find some simple and powerful parameters or scores to predict the development and to evaluate the risk of ARDS.

This study investigated the usefulness of ventilator parameters in the prediction of development and outcome of ARDS in mechanically ventilated patients on admission to the SICU. We found that a small value of MW/IER was the most important predictor of the development of ARDS, and large values of RPPv and RPVI were marginally significant predictors of mortality of ARDS patients. These results suggest that ventilator parameters can be used to predict the morbidity and mortality due to ARDS in postoperative patients with esophageal or lung cancer on admission to the SICU.

Only ventilator parameters were used in the statistical analysis of this study. Other clinical data, laboratory data, or scores such as APACHE II, SOFA score, CRP, WBC, and arterial blood gases were not included in the statistical analysis so as not to interfere with our original intention to test the usefulness of ventilator parameters in the prediction of development and outcome of ARDS in postoperative patients. If such ventilator parameters can be found, they can be used to alert the attending physician about the deteriorating condition and possible outcome of the patients. If the search for such ventilator parameters had included the above-mentioned data or scores or had been adjusted for the disease severity of the patients, then the clinical utility of the ventilator parameters would have been limited because their clinical utility would have relied on the data or scores of the patients, and such parameters would not be useful in patients with unknown disease severity or no relevant data.

The RSBI is thought to be a stress response reflecting the balance between the respiratory neuromuscular reserve and respiratory demands. It is a clinical parameter often used in the prediction of successful weaning from ventilators in ventilated patients [13,14,15,16]. The RSBI is obtained by measuring the respiratory rate and tidal volume of the patient without ventilatory support, while the RSBIv defined in this study was measured when the patient was still on the ventilator. Thus, the RSBIv is different from the RSBI used clinically to predict the weaning outcome of the patients, though the equations used to calculate the RSBI and RSBIv are the same. Because the lungs of the ARDS patients were more rigid than those of the control patients, their tidal volumes were smaller and their respiratory rates were greater, resulting in a greater RSBIv in ARDS patients than in control patients. That the RSBIv of non-surviving ARDS patients was greater than that of surviving ARDS patients can also be accounted for by the more rigid lung in the non-surviving ARDS patients. Although the mean value of RSBIv was significantly higher in ARDS patients as well as in non-surviving ARDS patients, the increase in RSBIv did not increase the risk of development of ARDS and the mortality risk of ARDS patients in the adjusted model. Though the RSBIv might be used to indicate the severity of lung involvement, it cannot be used to predict the development of ARDS and the mortality due to ARDS in mechanically ventilated patients on admission to the SICU.

The MW of the ARDS patients was found to be lower than that of the control patients in this study. This should not be interpreted as the lesser respiratory distress in the ARDS patients as compared to the control patients, because the MW measured the support delivered to the patients by the ventilators, which was not the same as the work of breathing measured in patients who are breathing on their own efforts. The lower MW in ARDS patients compared to control patients might be caused mainly by the smaller V_T_ due to edematous lungs in such patients. A small V_T_ can also lead to a shorter Te, resulting in a greater IER in ARDS patients as compared to the control patients.

By analogy with the RPP in cardiology, we define the RPPv to measure the stress imposed on the respiratory muscle in ventilated patients. In addition, by combining the RSBIV with RPPv, we can define the RPVI in ventilated patients. We found that both RPPv and RPVI of the ARDS patients were greater than those of the control patients. In ARDS patients, the greater RPPv was caused by the greater RR, and the greater RPVI was caused by the greater RR and the smaller V_T_. Moreover, we found that both RPPv and RPVI of the non-surviving ARDS patients were greater than those of the surviving ARDS patients. The greater RPVI in the non-surviving ARDS was caused by a greater PIP, and the greater RPVI in the same subgroup of patients was caused by a greater PIP and a smaller V_T_ in those ARDS patients. In the adjusted model, both RPPv and RPVI were marginally significant predictors of the mortality of ARDS patients. This result suggests that both RPPv and RPVI might have prognostic value in the management of ARDS patients.

The first and major limitation of this study was the small number of ARDS patients, which was due to the low incidence of ARDS in postoperative patients with lung or esophageal cancer. In order to resolve this issue, the PCA selection method, EPV-10 principle, and logistic regression with Firth’s penalized likelihood approach were chosen in this investigation so that the findings of this study could be substantiated. The second limitation was that this analysis was a retrospective one, so the study design could not be adjusted in accordance with the aim of the study. The third limitation was that no sequential ventilator data were collected and analyzed to see the changes in ventilator parameters during the course of ARDS. Further studies are needed to verify the findings of this study.

## 6. Conclusions

Some ventilator parameters can be defined and used to predict the development and outcome of the mechanically ventilated ARDS patients on admission to the SICU, such as RPPv, RPVI, and MW/IER defined in this study. A larger-scale study is needed to verify these findings and identify the most useful ventilator parameters in the prediction of development and outcome of ARDS patients.

## Figures and Tables

**Figure 1 diagnostics-11-00648-f001:**
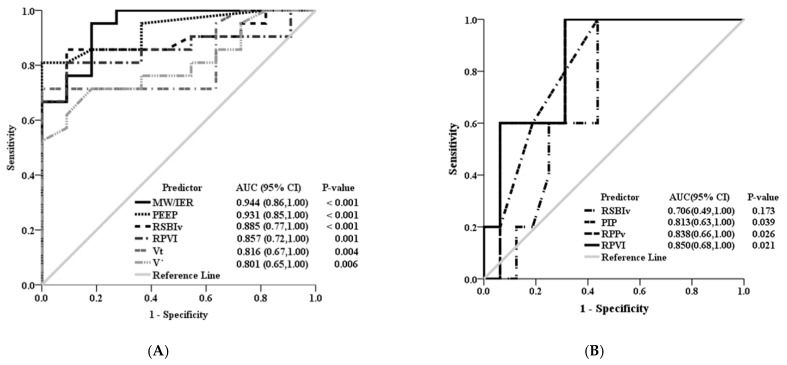
The areas under the receiver operating characteristic (ROC) curves (AUCs) and 95% confidence intervals (CIs) of mechanical ventilation parameters in predicting the development of ARDS and the mortality in patients. (**A**) The AUCs and 95% confidence intervals (CIs) of PEEP, RSBIv, RPVI, V_T_, and $\dot{V}$ in predicting the morbidity of ARDS patients were 0.931, 0.885, 0.857, 0.816, and 0.801, respectively. (**B**) The AUCs of RPVI, RPPv, RSBI, and PIP in predicting the mortality of ARDS patients were 0.850, 0.838, 0.706, and 0.813, respectively.

**Table 1 diagnostics-11-00648-t001:** Comparisons between non-ARDS and ARDS groups.

Variables	Non-ARDS (*n* = 11)	ARDS (*n* = 21)	*p*
General characteristics			
Age (years)	61 ± 8	60 ± 17	0.687
Gender (M/F)	5/6	17/4	0.056
BH (cm)	158.5 ± 10.3	164.7 ± 6.9	0.050
BW (kg)	65.6 ± 9.4	65.1 ± 8.5	0.882
BMI (kg/m^2^)	26.3 ± 4.4	24.0 ± 2.8	0.138
Clinical characteristics			
Lung cancer/esophageal cancer	6/5	9/5	0.697
SICU stay (days)	4 (2, 4)	25 (14, 36)	<0.001 *
Pneumonia (yes/no)	0/11	4/17	0.272
Hypertension (yes/no)	3/8	4/17	0.667
ASA classification	1 (1, 2)	2 (2, 3)	0.005 *
APACHE II	9 (6, 12)	16 (11, 19)	0.002 *
RASS	0 (0, 0)	−1 (−1, 0)	0.001 *
Ventilator parameters			
Endo size	7 (7, 7.5)	7.5 (7, 7.5)	0.134
PEEP (cmH_2_O)	5.8 ± 1.8	10.4 ± 3.0	<0.001 *
PIP (cmH_2_O)	20 (20, 20)	20 (18, 22)	0.983
RI (s)	5.0 (4.6, 5.0)	3.5 (2.3, 3.8)	<0.001 *
Ti (s)	0.9 (0.9, 1.0)	0.9 (0.8, 1.0)	0.532
Te (s)	3.8 ± 0.6	2.3 ± 0.9	<0.001 *
$\dot{V}$ (mL/s)	574.8 ± 92.5	464.7 ± 119.1	0.012 *
V_T_ (mL)	538.5 ± 93.8	427 ± 117.4	0.011 *
RR (bpm)	12 (12, 13)	17 (16, 26)	<0.001 *
MV (L/min)	7.1 ± 2.0	8.6 ± 3.2	0.160
Cdyn (mL/cmH_2_O)	28.6 ± 6.2	22.9 ± 7.6	0.041 *
Raw (cmH_2_O·s/mL)	0.034 ± 0.006	0.044 ± 0.012	0.020 *
RSBIv (bpm/L)	25.0 (20.3, 26.1)	56.3 (37.7, 69.6)	<0.001 *
RPPv (cmH_2_O * bpm)	240 (240, 260)	340 (300, 540)	0.010 *
RPVI (cmH_2_O * bpm/mL)	0.48 (0.39, 0.52)	0.98 (0.71, 1.35)	0.001 *
IER	0.24 (0.22, 0.25)	0.36 (0.32, 0.66)	<0.001 *
MW (cmH_2_O * L)	10.3 ± 2.1	8.2 ± 2.6	0.030 *
MW/IER (cmH_2_O * L)	41.1 (37.0, 43.7)	18.7 (12.8, 27.3)	<0.001 *
Outcome			
Survivors	11 (100%)	16 (76.2%)	0.138
Non-survivors	0 (0%)	5 (23.8%)	

ARDS: acute respiratory distress syndrome; ASA classification: American Society of Anesthesiologists Physical Status Classification System; APACHE II: Acute Physiology and Chronic Health Evaluation II; RASS: RASS: Richmond Agitation–Sedation Scale; bpm: breaths per min; PEEP: positive end-expiratory pressure; PIP: peak inspiratory pressure above PEEP; Ti: inspiration time; Te: expiratory time; V_T_: tidal volume; MV: minute ventilation; Cdyn: dynamic compliance; $\dot{V}$: flow rate; Raw: airway resistance; RSBIv: rapid shallow breathing index during mechanical ventilation; RPPv: rate pressure product of ventilation; RPVI: rate pressure volume index; MW: mechanical work; IER: inspiration to expiration time ratio. * *p* < 0.05.

**Table 2 diagnostics-11-00648-t002:** Comparisons between surviving and non-surviving patients with ARDS.

Variables	Survivors (*n* = 16)	Non-survivors (*n* = 5)	*p*
General characteristics			
Age (years)	57 ± 17	69 ± 12	0.148
Gender (M/F)	14/2	3/2	0.228
BH (cm)	165.3 ± 6.6	162.7 ± 8.3	0.493
BW (kg)	64.5 ± 7.6	67.2 ± 11.6	0.546
BMI (kg/m^2^)	23.62 ± 2.49	25.34 ± 3.49	0.235
Clinical characteristics			
Lung cancer/esophageal cancer	5/4	4/1	0.123
ICU stay (days)	20.5 (11, 40.5)	25 (25, 36)	0.509
Pneumonia (yes/no)	4/12	0/5	0.532
ASA classification	2 (2, 3)	2 (2, 3)	0.768
APACHE II	14 (10, 19)	18 (14, 29)	0.185
LIS	13 (10, 14)	13 (12, 15)	0.557
SOFA score	8 (7, 10)	7 (7, 8)	0.313
RASS	−1 (−1, 0)	−2 (−2, −1)	0.040 *
WBC (×10^3^/μL)	13.3 (7.7, 13.4)	12.9 (11.1, 14.0)	0.905
CRP (mg/dL)	21.5 (10.9, 22.6)	12.4 (11.0, 19.7)	0.495
Ventilator parameters			
Endo size	7.5 (7.0, 7.5)	7.0 (7.0, 7.5)	0.314
PEEP (cmH_2_O)	10.0 (9.0, 12.0)	10.2 (10.0, 11.9)	0.523
PIP (cmH_2_O)	18 (15.5, 20)	22 (20, 22)	0.035 *
RI (s)	3.41 ± 0.94	2.58 ± 0.66	0.087
Ti (s)	0.9 (0.8,1.0)	0.9 (0.8,0.9)	0.520
Te (s)	2.48 ± 0.88	1.70 ± 0.70	0.088
V_T_ (mL)	436.3 ± 119.5	397.2 ± 117.9	0.530
RR (breaths/min)	17 (16, 22)	26 (24, 27)	0.145
MV (L/min)	8.19 ± 2.84	9.96 ± 4.08	0.285
RSBIv (bpm/L)	47.8 ± 20.0	62.9 ± 10.4	0.045 *
RPPv (cmH_2_O * bpm)	359.5 ± 148.9	514.8 ± 92.7	0.042 *
RPVI (cmH_2_O * bpm/mL)	0.88 ± 0.39	1.35 ± 0.24	0.023 *
Cdyn (mL/cmH_2_O)	21.6 (18.9, 30.7)	19.0 (17.7, 21.3)	0.215
$\dot{V}$ (mL/s)	470.3 ± 122.1	446.9 ± 120.3	0.712
Raw (cmH_2_O·s/mL)	0.041 ± 0.010	0.051 ± 0.017	0.105
MW (cmH_2_O * L)	8.11 ± 2.72	8.46 ± 2.54	0.801
IER	0.35 (0.32, 0.45)	0.66 (0.60, 0.67)	0.230
MW/IER (cmH_2_O * L)	21.7 (12.2, 30.3)	12.9 (12.8, 18.1)	0.240

ARDS: acute respiratory distress syndrome; ASA classification: American Society of Anesthesiologists Physical Status Classification System; APACHE II: Acute Physiology and Chronic Health Evaluation II; LIS: lung injury score; SOFA: Sequential Organ Failure Assessment; RASS: RASS: Richmond Agitation–Sedation Scale; WBC: white blood cells; CRP: C-reactive protein; bpm: breaths per min; PEEP: positive end-expiratory pressure; PIP: peak inspiratory pressure above PEEP; Ti: inspiration time; Te: expiratory time; V_T_: tidal volume; MV: minute ventilation; Cdyn: dynamic compliance; $\dot{V}:$ flow rate; Raw: airway resistance; RSBIv: rapid shallow breathing index during mechanical ventilation; RPPv: rate pressure product of ventilation; RPVI: rate pressure volume index; MW: mechanical work; IER: inspiration to expiration time ratio. * *p* < 0.05.

**Table 3 diagnostics-11-00648-t003:** The areas under the ROC curves (AUCs) of mechanical ventilator parameters and variable coefficients in principal component analysis for the development of ARDS.

Variables	AUC Analysis	Principal Component Analysis	
		Component 1	Component 2
PEEP (cmH_2_O)	0.931	**0.760**	0.327
RSBIv (bpm/L)	0.885	**0.992**	0.041
RPVI (cmH_2_O * bpm/mL)	0.857	**0.949**	−0.091
MW/IER (cmH_2_O * L)	0.944	**0.877**	−0.025
RR (breaths/min)	0.881	**0.828**	−0.554
RI (s)	0.881	**0.828**	−0.554
Te (s)	0.907	**0.808**	−0.577
IER (per 0.1 increment)	0.905	**0.759**	−0.578
RPPv (cmH_2_O * bpm)	0.779	**0.743**	−0.587
Raw (cmH_2_O·s/mL)(per 0.01 increment)	0.788	**0.695**	0.576
Cdyn (mL/cmH_2_O)	0.784	**0.686**	0.511
$\dot{V}$ (mL/s)	0.801	0.603	**0.755**
MW (cmH_2_O * L)	0.639	0.504	**0.753**
V_T_ (mL)	0.816	0.677	**0.729**

ARDS: acute respiratory distress syndrome; bpm: breaths per min; PEEP: positive end-expiratory pressure; Te: expiratory time; V_T_: tidal volume; Cdyn: dynamic compliance; $\dot{V}$: flow rate; Raw: airway resistance; RSBIv: rapid shallow breathing index during mechanical ventilation; RPPv: rate pressure product of ventilation; RPVI: rate pressure volume index; MW: mechanical work; IER: inspiration to expiration time ratio. Variables with coefficients in bold font in the same column are included in the same component.

**Table 4 diagnostics-11-00648-t004:** Firth logistic regression analyses with stepwise elimination procedure for predicting the development of ARDS and mortality in patients with ARDS.

**Significant Predictors of the Development of ARDS**						
**Variables**	**Model 1a**		**Model 2a**		**Model 3a**	
	**Adj. OR**	***p***	**Adj. OR**	***p***	**Adj. OR**	***p***
MW/IER	0.82 (0.71, 0.94)	0.004 *			0.81 (0.70, 0.95)	0.008 *
V_T_			0.99 (0.98, 1.00)	0.033 *	1.00 (0.99, 1.01)	0.651
**Predictors of Mortality in Patients with ARDS**						
**Variables**	**Model 1b**		**Model 2b**		**Model 3b**	
	**Adj. OR**	***p***	**Adj. OR**	***p***	**Adj. OR**	***p***
PIP	1.35 (0.90, 2.02)	0.150				
RSBIv	1.05 (0.98, 1.13)	0.190				
RPPv			1.01 (1.00, 1.02)	0.090		
RPVI					28.4 (0.8, 1034.8)	0.068

ARDS: acute respiratory distress syndrome; Adj. OR: adjusted odds ratio; V_T_: tidal volume; MV: minute ventilation; $\dot{V}$: flow rate; RSBI: rapid shallow breathing index during mechanical ventilation; RPPv: rate pressure product of ventilation; RPVI: rate pressure volume index; MW: mechanical work. * *p* < 0.05. Adjusted OR was calculated by multivariate logistic regression with stepwise elimination procedure. The first step of model 1a was adjusted for RSBIv, RPVI, and MW/IER. The first step of model 2a was adjusted for $\dot{V}$, MW, and V_T_. Model 3a was adjusted for MW/IER and V_T_. Model 1b was adjusted for PIP and RSBIv. Model 2b was adjusted for RPPv. Model 3b was adjusted for RPVI.

## Data Availability

The datasets used and/or analyzed during the current study are available from the corresponding author on request.

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
