# Peer review of "Ventilator Parameters in the Diagnosis and Prognosis of Acute Respiratory Distress Syndrome in Postoperative Patients: A Preliminary Study"

_diagnostics, 2021, doi:10.3390/diagnostics11040648_

Round 1
Reviewer 1 Report
The authors made the required adjustments. Authors have also underlined the limits of this study in the final part of the discussion
Author Response
Thank you very much for your precious comments and suggestions. Best regards.
Reviewer 2 Report
I am not satisfied by 2 of your answers:
1) I would like a better answer and this should also be included in your manuscript.
Re. Did any of the patients develop sepsis?
Ans: No patients in the control group had pneumonia, while 4 patients in the ARDS group had pneumonia which might lead to certain degree of sepsis. However, none of the ARDS
patients with pneumonia expired at the end of this study. Therefore, sepsis might not be a problem in this study.
I did not ask specifically about pneumonia. I asked about sepsis. Also, there is no "degree" of sepsis. They either developed sepsis or not. ARDS is usually sepsis-induced.
Please clarify.
2) Re. Table 1 is missing a lot of significant parameters, including PCT, CRP, APACHE II and SOFA scores. These should be included in the analyses.
Ans: The APACHE II, SOFA score, CRP, WBC and other clinical relevant data had been presented in our previous article. Some of them are presented in this manuscript again for
the sake of convenience.
So because they were recorded in a previous publication, they are not being used in your current analysis? How can you perform MLR without adjusting for disease severity?
3) N=5 for non-survivors is way too small to do any statistics. ROC curves require at least 10 events.
Also, MLR you state in the Methods: However, only three variables can be included in the multivariable analysis based on the rule of 10 events per variable (EPV-10).
You only have 21 cases. The results of the MLR cannot be used. They are totally biased, since you had to choose which parameters to include.
Author Response
Thank you very much for your comments and suggestions
Re. 1) I would like a better answer and this should also be included in your manuscript.
Re. Did any of the patients develop sepsis?
Ans: No patients in the control group had pneumonia, while 4 patients in the ARDS group had pneumonia which might lead to certain degree of sepsis. However, none of the ARDS
patients with pneumonia expired at the end of this study. Therefore, sepsis might not be a problem in this study.
I did not ask specifically about pneumonia. I asked about sepsis. Also, there is no "degree" of sepsis. They either developed sepsis or not. ARDS is usually sepsis-induced.
Please clarify.
Ans2: Thank you very much for bringing up this question. The diagnosis of many clinical diseases including sepsis and ARDS are often not very easy to be definite and clear-cut in clinical settings. Many scores and diagnosis criteria are then devised and developed to assist the attending physicians in their management and treatment of clinical illness. Therefore, I used the term “certain degree of sepsis” in my previous response to your comments to mean that many clinical diseases are often not very easy to be diagnosed with 100% certainty. Since no laboratory data associated with sepsis were available for those ARDS patients except WBC and CRP, the patients might not have signs of sepsis so that the attending physician had not done the necessary laboratory examinations of sepsis for them. Thus, we believe that none of our ARDS patients had clinically manifest sepsis.
Re. 2) Re. Table 1 is missing a lot of significant parameters, including PCT, CRP, APACHE II and SOFA scores. These should be included in the analyses.
Ans: The APACHE II, SOFA score, CRP, WBC and other clinical relevant data had been presented in our previous article. Some of them are presented in this manuscript again for the sake of convenience.
So because they were recorded in a previous publication, they are not being used in your current analysis? How can you perform MLR without adjusting for disease severity?
Ans2: This study intended to investigate the usefulness of “ventilator parameters” alone in the prediction of development and outcome of acute respiratory distress syndrome (ARDS) in postoperative patients with esophageal or lung cancer on admission to the surgical intensive care unit (SICU). Only ventilator parameters were used in the statistical analysis of this study. Other clinical data, laboratory data or score such as APACHE II, SOFA score, CRP, WBC, and arterial blood gases were not included in the statistical analysis so as not to interfere with our original intention to test the usefulness of “ventilator parameters” in the prediction of development and outcome of ARDS in postoperative patients. If such ventilator parameters can be found, they can be used to alert the attending physician about the deteriorating condition and possible outcome of the patients. If the searching of such ventilator parameters included the above-mentioned data or score, or were adjusted for the disease severity of the patients, then the clinical utility of those ventilator parameters will be limited because their clinical utility relies on the data or score of the patients and cannot be used in patients with unknown disease severity or have no relevant data.
In this study, the APACHE II, ALI score and SOFA score were not significantly different between survivors and non-survivors of ARDS, while the RSBIv, RPPv and RPVI were significantly elevated in the non-survivors of ARDS. It seems that the curently used APACHE II, ALI score and SOFA score in critical care have no capability to predict the mortality in patients with ARDS, but the RSBIv, RPPv and RPVI might have. By using PCA, both RPPv and RPVI were found to be the significant predictors of mortality in patients with ARDS. Hence, it is not necessary to include the APACHE II, ALI score and SOFA score in the searching of ventilator parameters for the prediction of outcome in ARDS patients.
Re. 3) N=5 for non-survivors is way too small to do any statistics. ROC curves require at least 10 events.
Also, MLR you state in the Methods: However, only three variables can be included in the multivariable analysis based on the rule of 10 events per variable (EPV-10).
You only have 21 cases. The results of the MLR cannot be used. They are totally biased, since you had to choose which parameters to include.
Ans2: We thank reviewer’s suggestions. The Medcalc software was used to calculate the sample size for ROC. The AUC of RPVI in the prediction of mortality in ARDS patients was 0.850 with 70% statistical power when the sample size was 16 cases for survival and 5 cases for non-survival. If we want to have a power of 80% and a ROC of 0.850, then 7 cases will be requied for survival and 23 cases for non-survival. Therefore, the ROC curve requires at least 10 events, which may not be necessary.
For the prediction of the development of ARDS and the mortality in ARDS patients, we conducted two MLR analyses. The first MLR model was used to predict the development of ARDS. We have 33 cases and three variables (RSBIv, RPVI, and MW/IER) which were included in the multivariable analysis. After performing stepwise elimination procedure, the MW/IER or VT remain in the model. A similar procedure was used for the second model to predict the mortality in ARDS patients. We have 21 cases, and two variables (PIP and RSBIv) were included in the multivariable analysis. Even if stepwise elimination procedure was performed, none of these two variables significantly affect the prediction of mortality in ARDS patients. However, the RPPv or RPVI had marginal effect in the prediction of mortality in ARDS patients in univariate analysis.

Round 2
Reviewer 2 Report
I thank the Reviewers for their answers.
1. Please allow me to disagree ony our first answer, where you mention that "The diagnosis of many clinical diseases including sepsis and ARDS are often not very easy to be definite and clear-cut in clinical settings." What about the established Sepsis-3 and Berlin definitions. So I ask again, how many patients had sepsis?
2. What is the "ALI score" you mention?
3. You conclude that : "It seems that the curently used APACHE II, ALI score and SOFA score in critical care have no capability to predict the mortality in patients with ARDS".
In your very limited cohort. You cannot draw such conclusions with such a small sample size. Please delete. It is also beyond the scope of your study.
Author Response
Re. 1. Please allow me to disagree ony our first answer, where you mention that "The diagnosis of many clinical diseases including sepsis and ARDS are often not very easy to be definite and clear-cut in clinical settings." What about the established Sepsis-3 and Berlin definitions. So I ask again, how many patients had sepsis?
Ans: Thank you very much for your precious comments. As has been stated in our previous response, the clinical and laboratory data associated with sepsis were not available except WBC and CRP level, suggesting that our ARDS patients might not have clinically manifest sepsis. Therefore, we believe that none of our ARDS patients had sepsis. The ARDS has over 60 etiologies. The common causes of ARDS include sepsis, aspiration pneumonia, infectious pneumonia, severe trauma + multiple fracture, pulmonary contusion, inhalation injury, massive blood transfusion, pancreatitis, thoracic surgery, certain drugs, etc. If we ask how many patients had sepsis, it might be necessary to ask how many patients had aspiration pneumonia, infectious pneumonia, inhalation injury, or massive blood transfusion, etc., for the sake of completeness. This surely is not possible for a small-scale study like ours. In this preliminary study, stratification of the ARDS patients into two or more subgroups based on their clinical etiologies might lead to invalid statistical analysis. Moreover, this study intended to find the usefulness of ventilator parameters alone in the diagnosis and prognosis of ARDS in post-operative thoracic cancer patients on admission to the SICU regardless of their etiology and other clinical conditions. If we include the etiologies of ARDS, scores, biochemical parametes of the patients in the statistical analysis, the usefulness of ventilator parameters in the diagnosis and prognosis of ARDS will be weakended and even invalidated. Therefore, whether or not the ARDS patients had sepsis, aspirtion pneumonia, etc., were not taken into statistical analysis in this study.
Re. 2. What is the "ALI score" you mention?
Ans: Acute lung injury score. The full name of “ALI score” has been presented in the legend of Table 2.
Re. 3. You conclude that: "It seems that the curently used APACHE II, ALI score and SOFA score in critical care have no capability to predict the mortality in patients with ARDS". In your very limited cohort. You cannot draw such conclusions with such a small sample size. Please delete. It is also beyond the scope of your study.
Ans: This statement is meant to represent the finding “in this study” only, not to mean it in the general case. Since only ventilator parameters were used in the statistical analysis in this study, this paragraph regarding the APACHE II, ALI score, and SOFA score of the ARDS patients is removed to avoid misunderstanding.

Round 3
Reviewer 2 Report
I thank the authors for their response. Only one minor comment. Please correct ALI score. It is LIS (lung injury score). The term ALI no longer exists, it has been replaced by ARDS.
Author Response
Re. I thank the authors for their response. Only one minor comment. Please correct ALI score. It is LIS (lung injury score). The term ALI no longer exists, it has been replaced by ARDS.
Ans: Many thanks for reviewing our manuscript in details to improve it greatly, and for telling me that the ALI score has been replaced by LIS (lung injury score). The ALI score in the manuscript has been changed to LIS in the revised manuscript.
This manuscript is a resubmission of an earlier submission. The following is a list of the peer review reports and author responses from that submission.
Round 1
Reviewer 1 Report
Many thanks for the opportunity to review the manuscript entitled “Ventilator Parameters in the Diagnosis and Prognosis of Acute Respiratory Distress Syndrome in Postoperative Patients: A preliminary Study”.The manuscript retrospectively evaluated ventilator parameters in the prediction of development and outcome of acute respiratory distress syndrome (ARDS) in 32 postoperative patients with esophageal or lung cancer on admission to the surgical intensive care unit. Some data of patients characteristics (as the number of patients with esophageal cancer and number of patients with lung cancer) have already been published by the authors: Chen, I.C.; Kor, C.T.; Lin, C.H.; Kuo, J.; Tsai, J.Z.; Ko, W.J.; Kuo, C.D. High-frequency power of heart rate variability can predict the outcome of thoracic surgical patients with acute respiratory distress syndrome on admission to the intensive care unit: a prospective, single-centric, case-controlled study. BMC Anesthesiol 2018, 18, 34. However, in this manuscript the authors explored a different issue. The manuscript is well written and organized. I suggest only few corrections:
- In the method section: This was a prospective, single-centric, case-controlled study with retrospective data analysis. Please delete prospective as authors made a retrospective data analysis
In the table 1 instead of control use non –ARDS
- In the table 1 and table 2 I suggest to report also how many patients with esophageal and lung cancer have authors included in the study in the non –ARDS group and how many in the ARDS group (these data were already reported in the published article) and to report and analyze them also in the survivors and non survivors and I suggest also to consider other parameters as other patients comorbidities as: how many have also pneumonia, how many an immunosuppression state and how many showed endothelial and coagulation disorders as authors have excluded patients with autoimmune system and patients with arrhythmia, cardiac pacing, diabetes mellitus, cerebral vascular accident, or major diseases of kidney. Please include also some laboratory parameters as the white blood count, the lymphocytes, the CRP and the D-Dimer level
- Please edit the references number 1-10 because they haven’t the number, other the DOI in your references and edit them on the base of the Journal’s style

Reviewer 2 Report
This study investigated the usefulness of ventilator parameters in the prediction of development and outcome of acute respiratory distress syndrome (ARDS) in postoperative patients with esophageal or lung cancer on admission to the surgical intensive care unit (SICU). Among 32 patients, 21 and 11 patients were classified as ARDS and non-ARDS group, respectively. They identified several parameter derived from mechanical ventilator can help predict the risk and the prognosis of ARDS. Although this study is interesting, I have several concerns about this study.
- In the abstract section, there was no any solid data to support your conclusions.
- It is difficult to understand why the study site was National Taiwan University Hospital, but you need to obtain the IRB from Changhua Christian Hospital and Taipei Veterans General Hospital.
- In addition, if this is a prospective study with retrospective data, you need to obtain patient consent.
- Please briefly describe the selection criteria.
- Please add more clinical characteristics, such as the cause of ARDS and underlying disease in the table 1.
- Please also briefly describe the cause of your limitation.
Reviewer 3 Report
In this manuscript, the Authors conclude that some ventilator parameters can be derived from ventilator readings and be used to predict the development and outcome of ARDS in mechanically ventilated patients. However, it was a retrospective analysis of the data to find the key ventilator parameters for the prediction of development and outcome of ARDS.
I am a bit confused on the methodology. So prospectively, all patients who met the inclusion criteria were enrolled, the ventilator readings were taken, and the patients were eventually grouped into ARDS and non, whether or not they developed ARDS during their ICU stay.
Were the ventilator settings the same throughout their stay?
Table 1 is missing a lot of significant parameters, including PCT, CRP, APACHE II and SOFA scores. These should be included in the analyses.
Did any of the patients develop sepsis?